# *Glycine soja*, PI424025, is a valuable genetic resource to improve soybean seed-protein content and composition

**Earl Taliercio[1]\*, Jay Gillenwater[2], Lisa Woodruff[3], Ben Fallen[1]**

1 Soybean and N Fixation Research Unit, USDA/ARS, Raleigh, North Carolina, United States of America,
2 Soil and Crop Science, North Carolina State University, Raleigh, North Carolina, United States of America,
3 Department of Crop and Soil Sciences, University of Georgia, Athens, Georgia, United States of America

\* Earl.Taliercio@usda.gov

**Data Availability Statement:** We provide the data as supplementary data in this manuscript.

## Abstract

Soybean seed-protein content and composition is important because it contributes over half the value of this $61 billion crop. Historic negative correlation between seed-protein content and oil have been reported. Similarly, negative correlation between seed-protein content and yield have been reported but may be at least in part mitigated by increasing the genetic diversity of the elite soybean germplasm. Improvements in amino acid composition of seed-protein would increase the value of soybean meal and protein for animal and human consumption. We have identified a genetic resource in wild soybean germplasm, PI424025B, with elevated seed-protein, elevated cysteine in the seed and increased sulfur content in the seed. We have developed a population of Recombinant Inbred Lines derived from a cross between NC-Raleigh and PI424025B. We evaluated the carbon, nitrogen and sulfur (C, N and S) content of seeds from the progeny, the parents and other high-seed protein soybean lines. N content and C/N (ratio of C to N) were well correlated with protein content measured by NIR. PI424025B had a N content comparable to high-protein soybean lines and a superior N/S. These traits were inherited by some of the progeny. Quantitative Trait Loci (QTL) associated with high-seed protein were identified on chr2, chr20 and chr15 and colocalizes with well characterized loci on chr 15 and chr 20 known to affect seed-protein content and quality. A potentially unique QTL was identified on Chr15. Unlike the chr20 QTL, the chr15 QTL improved S content relative to N content and was superior to other high seed-protein phenotypes tested. These data indicate that PI424025B is a valuable resource to diversify the genetics of soybean while improving soybean seed-protein content and composition.

## Introduction

Protein content has been well studied in soybean seeds because much of the value of this important crop is derived from its seed-protein content. The U.S. soybean crop was worth over $61 billion in 2022 and protein accounted for just over half of the value (Soy.xlsx (live. com)). The soybean crop is also an important source of oil which accounted for just under half of the crop's value in 2022 (Soy.xlsx (live.com)). Substantial effort had been aimed at

**Funding:** This research was funded by the United Soybean Board project number 2333-203-0101 and USDA ARS (6070-21220-069-000D).

**Competing interests:** The authors have declared that no competing interests exist.

identifying the genetics of inheritance of seed-protein quantity and quality in soybean [1–12]. There is a well-established negative correlation between quantity of seed protein and quantity of seed oil and between seed protein and yield [13–15]. These historic correlations may be due in part to the narrow genetic base of soybean in the U.S. Recent germplasm releases have identified genetic resources that at least in part mitigate these negative associations [16–18]. For example, USDA-N7007 is a high-seed protein line with wild ancestry that also yields well [19]. There are novel genetic resources in the *G. max* germplasm that improve protein composition. For example, the "Danbaekkong" allele is associated with a 2% increase in protein when back-crossed into elite lines but a substantial decrease in oil and yield [20]. However, it may be possible to select high yielding, high seed-protein lines from lines with the Danbaekkong allele [11]. Studies indicate that BARC-7 phenotype is associated with increased seed-protein content but is distinct from the Danbaekkong allele [10]. However, the negative correlation between seed-protein and seed oil persists.

The genetic inheritance of seed-protein content has been investigated and 16 confirmed QTL on 8 chromosomes are reported in Soybase from biparental mapping populations [4, 5, 21, 22]. It is likely that some of these loci on the same chromosome represent the same QTL. Multiple studies have localized high seed-protein QTL to chr20 (20–40 cM) and chr15, (30–40 cM), that impacts seed-protein content. In both cases the gene associated with the elevated seed-protein phenotype has been identified. The Danbaekkong allele is one of many high-seed protein QTL that maps to chr20. Variants of POWR1(Glyma.20g085100), a CONSTANS-like gene on chr20, affects the seed-protein content. Lower expression of the intact gene is associated with increased seed-protein content. A seminal finding was that the gene in *G. soja* is a functional copy and the gene in *G. max* has insertion that disrupts nuclear localization [13]. The GMSWEET39 (Glyma.15G049200) gene is a sugar transporter on chr15 affecting seed-protein content, seed amino acid composition and seed-oil content associated with that QTL [23].

Feed for animals is the most important market for soy meal (https://mosoy.org/about-soybeans/soybean-uses/soybeans-as-feed/). While an excellent source of protein, soymeal is low in the sulfur-containing amino acids [24]. Optimizing the sulfur-containing amino acid content of soy protein would add value to soy protein for human consumption and for animal feed [24, 25]. Soybase reports 33 QTL on 13 chromosomes that control content of S-containing amino acids in soybean seeds, though none appear to be confirmed QTL [21]. The two QTL (near markers BARC-066103-17539 and Sat_273) associated with S-containing amino acid content on chr15 are 10 Mbp and 30 Mbp from the SWEET39 gene making it unlikely they are the same loci. There are a cluster of markers (BARC-026051-05237, BARC-039753-07565, and BARC-042897-08454) on chr20 associated with variation in S-containing amino acid content in soybean seeds. Whether these represent the same QTL is ambiguous, but it is likely that at least one overlaps the POWR1 locus.

Phenotyping the *G. soja* germplasm identified a wild accession, PI424025B, with high-seed-protein, slightly elevated cysteine in seeds and elevated sulfur content in seeds [26, 27]. We have used a population of Recombinant Inbred Lines (RIL) derived from a cross between NC-Raleigh and PI424025B to explore the high seed sulfur phenotype derived from PI424025B.

## Materials and methods

### Plant material

NC-Raleigh and PI424025B were crossed in the summer of 2014 at Central Crops Research station (CCRS) in Clayton, North Carolina [28]. NC-Raleigh was chosen as a parent because

its flowering period overlapped with PI424025B, it is lodging resistant and is resistant to frog-eye leaf spot. The $F_1$ seeds were grown in the Winter Nursery in Puerto Rico. $F_2$ seeds grown in CCRS were used to found the mapping population. One seed was harvested from each plant. The population was advanced to the $F_4$ generation by planting one seed per plant of each generation in the greenhouse between 2015–2016.

The mapping population was increased in the spring of 2017 in the greenhouse by harvesting seeds from 151 individual $F_5$ progeny and the parents. Near-maturity plants were placed in an open mesh bag to facilitate harvest of seeds from plants with pods that shattered. In 2017, seeds were prepared for evaluation in two locations in NC, one location in SC, one location in MS and one location in LA. The seeds of each progeny were randomized among plots in a Randomized Complete Block Design (RCBD) of two replications. The odd plots were the mapping population, and the even plots were the soybean variety Stressland. Planting Stressland in even plots created a checkerboard of plants that separated the vine-like progeny from each other in the field to prevent intermingling of the plants. Stressland is a maturity group three cultivar that exhibits a short plant height when grown in the southern U.S. Seeds were planted in five-foot single row plots separated by two-foot alleys. These progenies shatter, so plants were harvested in paper bags and allowed to shatter before harvesting in a belt thresher. To harvest sufficient seeds for NIR analysis plants grown in the Jackson Springs research station (JSRS) were put into a large mesh bag when shattering was observed and kept in the bag until most of the pods had shattered.

## NIR and analyses of C, N and S

Seeds from all locations were hand cleaned and ground using the 2mm screen on a Retsch ZM (Retsch, Haan Germany) 100 centrifugal grinder. Care was taken to avoid cross contamination by scrupulously cleaning between each sample. NIR analysis to measure protein and oil content was performed on the samples from JRS in the large dish (108 cm$^2$ surface area) following standard protocols on the Perten NIR DA7250 (PerkinElmer, Shelton CT). For elemental analysis of carbon, nitrogen, and sulfur on the FlashEA Elemental Analyzer (ThermoFisher, Waltham MA) the ground samples were dried to a constant weight at 61°C. Three to six mg of dried, ground seed was added to a tin cup that included nine to eleven mg of Vanadium Oxide. Vanadium Oxide is required to measure sulfur. Soybean leaves purchased from the vendor were used as the standard and a high-seed protein (PI424025B) and a low-seed protein (NC-Raleigh) standard were included in every 60 samples to monitor the measurements. Samples that were outliers (more than 2 standard deviations) were redone. The elemental analyzer data is available in S1 Data.

## Molecular marker and QTL analysis

Genomic DNA was extracted from the first trifoliate of greenhouse-grown plants in 2017 with a Qiagen DNAeasy™ plant mini kit (Qiagen) following the manufacturers protocol. DNA was sent for genotyping at the USDA-ARS Soybean Genomics & Improvement laboratory at Beltsville, MD. Genomic DNA was genotyped with 6000 SNP markers from the BARCSoySNP6K Beadchip (6K Chip) [29, 30]. Marker data is available in S2 Data. To identify genes in the region of QTL, we identified the sequences of the right and left markers (shown in S3 Data) in the Wm82.a2 assembly using the BLAST tool at soybase [21]. The tool in the soybase genome browser was used to download the gene models in the targeted region and the soybase annotation tool was used to download the annotation.

## Phenotype data analysis

Histograms of phenotypic data were produced with the ggplot2 package with the R statistical software and were inspected for normality [31]. Potential deviations from normality were further inspected by calculating the skewness and kurtosis of each phenotypic distribution with the moments package in R [32]. Phenotypes were observed to be generally normally distributed with no obvious skew or multimodality based on these plots and statistics. Principal Component analysis was performed in R using prcomp, data was centered and scaled. The biplot was generated using the factorextra package (1.0.7) biplot function. Missing data was omitted except for the data for NC-Raleigh grown in SC. The average NC-Raleigh data were used for SC so that the point would be present on the biplot. ANOVA was performed in R using the car package [33].

## Linkage map construction

Genotypes were inspected for the presence of likely genetic clones and only one genotype from each pair of genotypes which shared greater than or equal to 90% of their alleles in common was retained. Any genotype with more than 10% missing marker data was removed. Markers were removed if they were monomorphic or were missing in either parent. Further marker quality control was performed with the ASMap package in the R statistical software [34]. Marker quality control was performed with the *pullCross* function from ASMap to remove markers based on the following quality criteria. The *pullCross* function filtered Co-located markers to retain only one marker from sets of markers with identical segregation patterns and removed markers if they had a p-value in a chi-square segregation distortion test less than 0.001. Markers with more than 5% missing data were removed. Following the application of these filtering criteria, the genetic map was constructed using the *mstmap.cross* function from the ASMap package using default arguments. Markers generally clustered into linkage groups corresponding to the chromosomal assignments of the SNP markers, and closely matched the physical order of the markers. The 17 markers which did not group to expected linkage groups were removed from the linkage map. Pairwise recombination fractions of the ordered markers were visualized with the *plotRF* function from the qtl package [35]. The visualization of recombination fractions did not reveal any obvious issues with marker orders or groups. The final linkage map contained 1813 SNP markers for 142 soybean genotypes. The final linkage map had a length of 2942 cM with an average marker spacing of 1.6 cM and a maximum marker spacing of 17.7 cM. The Kosambi mapping function was used to estimate map distances [36].

## QTL mapping

QTL were mapped using multiple interval mapping with the QTL package in R. Multiple QTL model search was performed with the *stepwiseqtl* function with the Haley-Knott regression method and penalties derived from 2500 permutations of the mapping data [35, 37]. Model statistics were obtained from QTL models with the *fitqtl* function, again from the qtl package. Phenotype by genotype distributions were visualized at the sites of markers closest to detected QTL with the *plotPXG* function from the qtl package. Linkage maps and QTL positions were visualized with the LinkageMapView package in R [38].

## Results

Screening of the wild soybean germplasm identified accessions that were higher in seed protein and had superior amounts of seed sulfur [26]. One of these accessions, PI424025B, had over 48% seed protein and modest statistically significant increases in cysteine content when

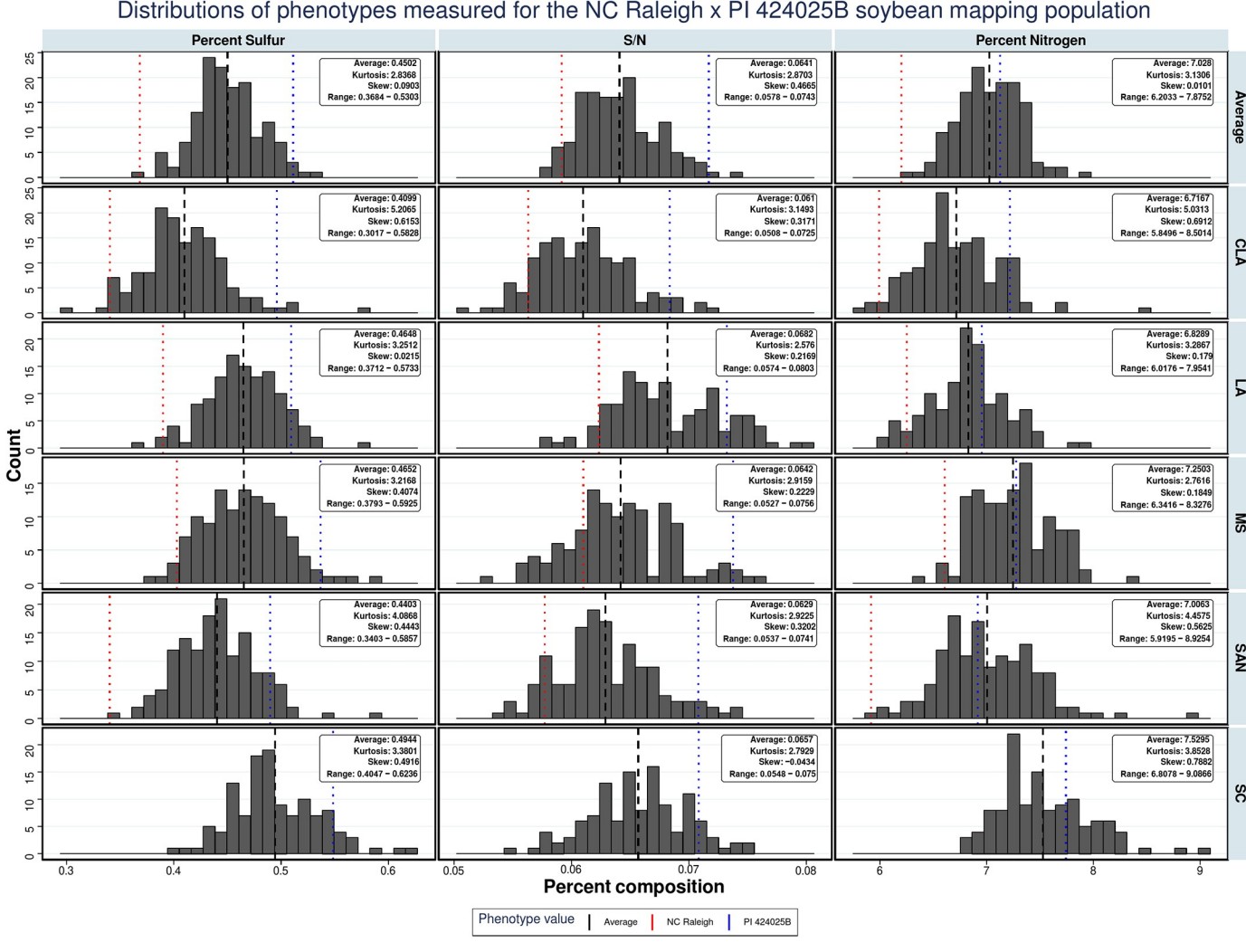

**Fig 1. The distribution of phenotypes if NC-Raleigh, PI424025B and their progenies.** The average, kurtosis, skew and Range are reported.

grown in NC environments. A biparental mapping population, created from a cross between PI424025B and a high-yielding domesticated soybean, NC-Raleigh, was grown in five diverse environments in four southern states to identify QTL associated with elevated seed-protein and seed-sulfur content from PI424025B. Seed amino acid content was not measured due to the cost of the assay. Carbon, nitrogen and sulfur were measured in dry soybean meal from the parents and progenies. Also included were ratios of N to C (N/C), S to C (S/C) and nitrogen to sulfur (N/S). Ratios are often included in data from elemental analyses [39–41]. Significant correlation of shoot N/C with QTL were identified in a panel of soybean lines [39]. Fig 1 shows the distribution of phenotypes in all environments and averaged across environments. The near-normal distribution of these values allows the mapping of loci affecting soybean seed composition.

Fig 2 shows the principal component analysis of the % C, %S, %N, N/C, S/C and N/S from all locations and the average of locations. Dimension 1 and dimension 2 account for a variation of 41% and 15% respectively. The variation of these values is similar across locations. N and N/C contribute substantially to variation and are inversely correlated to %C which lies

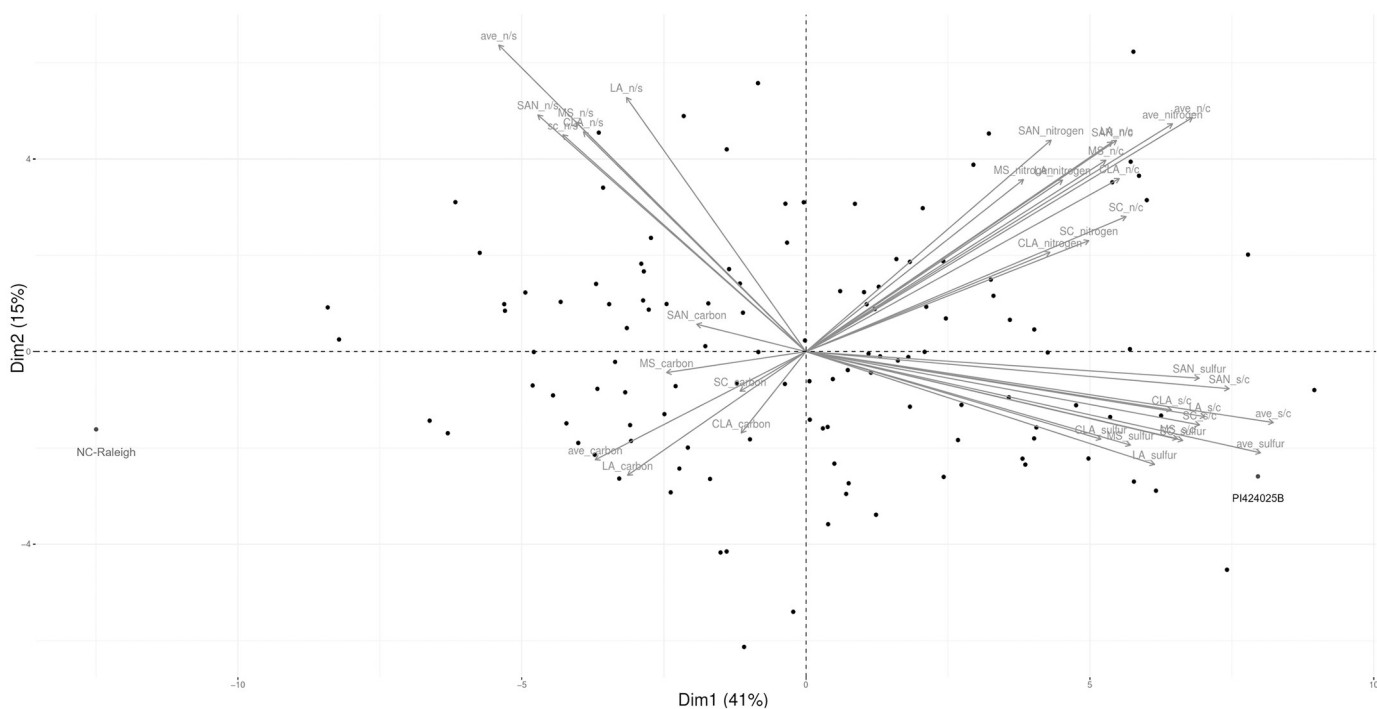

**Fig 2. The principal component analysis of the % C, %S, %N, N/C, S/C and N/S from all locations and the average of locations.** Dimension 1 and dimension 2 account for a variation of 41% and 15% respectively.

along the same line but in the opposite direction. N/S contributes to DIM2 and interestingly is not closely correlated with %N or S or S/C.

ANOVA analysis of phenotypic data is shown in Table 1. There is a substantial Environment and GenotypeXEvirionment affect for all phenotypes except the genotypeXEnvirioment effect of %C is insignificant. The broad sense heritability ($H^2$) for %S and S/N is low at 0.22 and 0.34 respectively. However, the $H^2$ for N/S is high at 0.89. Broad sense heritability for %N and N/C is 0.73 and 0.85, respectively.

These progenies are 50% wild soybean and suffer from many of the same poor agronomic qualities of the wild parent. For example, many shatter badly and the plant architecture is vine-like making machine harvesting impractical. Therefore, it was impractical to evaluate some agronomic traits like yield. Progenies were grown in small plots, resulting in insufficient seeds for NIR from most environments. An advantage of measuring C, N and S in an elemental analyzer is that only small amounts of seeds are needed. It is possible to estimate protein content from N content [42]. Typically an N to protein conversion factor of 6.25 is used but there is debate about the best value to use [42].Two replications from a North Carolina (Sandhills) location had enough seeds to attempt measuring protein and oil with NIR. There was a correlation between protein and N content (0.70) as expected confirming that these data are relevant to commonly used NIR measurements. The ratio of N to C gave a better correlation of (0.78) with seed protein measured by NIR. There was a negative correlation (-0.8) between protein and oil confirming the well reported negative correlation between seed-protein and seed-oil for this population.

We also evaluated C, N and S content of these parents and selected progeny relative to other high protein soybean lines (Fig 3) [11, 43]. The progeny high and low in protein were

**Table 1. ANOVA of %N, %C, %S and their ratios.**

| Phenotype | Source | Sum Sq | Df | F value | Pr(>F) | H$^2$ |
|---|---|---|---|---|---|---|
| %N | Genotype | 95.5945 | 150 | 4.9 | 2.08E-45 | 0.73 |
| | Environment | 103.9836 | 4 | 199.94 | 1.36E-110 | |
| | Genotype:Environment | 109.6940 | 536 | 1.57 | 2.27E-08 | |
| | Environment:Rep | 3.9257 | 5 | 6.04 | 1.80E-05 | |
| | Residuals | 81.9107 | 630 | - | - | |
| %C | Genotype | 968.8116 | 150 | 1.84 | 2.30E-07 | 0.48 |
| | Environment | 703.2917 | 4 | 50.01 | 1.45E-36 | |
| | Genotype:Environment | 2057.3504 | 536 | 1.09 | 0.144734 | |
| | Environment:Rep | 166.4314 | 5 | 9.47 | 1.02E-08 | |
| | Residuals | 2214.9130 | 630 | - | - | |
| %S | Genotype | 1.1220 | 150 | 8.68 | 9.40E-86 | 0.22 |
| | Environment | 0.9926 | 4 | 287.97 | 1.19E-140 | |
| | Genotype:Environment | 0.7360 | 536 | 1.59 | 9.68E-09 | |
| | Environment:Rep | 0.0527 | 5 | 12.23 | 2.46E-11 | |
| | Residuals | 0.5429 | 630 | - | - | |
| N/S | Genotype | 808.3142 | 150 | 7.44 | 1.52E-73 | 0.89 |
| | Environment | 454.5552 | 4 | 156.8 | 4.75E-93 | |
| | Genotype:Environment | 598.4761 | 536 | 1.54 | 9.38E-08 | |
| | Environment:Rep | 35.2342 | 5 | 9.72 | 5.83E-09 | |
| | Residuals | 456.5703 | 630 | - | - | |
| N/C | Genotype | 0.0484 | 150 | 10.37 | 7.13E-101 | 0.85 |
| | Environment | 0.0219 | 4 | 175.86 | 4.47E-101 | |
| | Genotype:Environment | 0.0283 | 536 | 1.69 | 1.03E-10 | |
| | Environment:Rep | 0.0006 | 5 | 4.05 | 0.001271 | |
| | Residuals | 0.0196 | 630 | - | - | |
| S/C | Genotype | 0.0005 | 150 | 13.32 | 5.65E-124 | 0.34 |
| | Environment | 0.0003 | 4 | 283.98 | 2.02E-139 | |
| | Genotype:Environment | 0.0002 | 536 | 1.56 | 4.08E-08 | |
| | Environment:Rep | 0.0000 | 5 | 10.07 | 2.72E-09 | |
| | Residuals | 0.0002 | 630 | - | - | |

grown in two reps in three environments in North Carolina. Included was a high-seed protein line with the BARC7 phenotype, cultivar Benning HP which contains the Danbaekkong allele and a recurrent cross with the low-seed protein allele at the chr20 locus [10, 11, 43]. The parents of the mapping population had distinctly different levels of C, N and S. NC-Raleigh was higher in C content than PI424025B but some progeny, like 1923, fell below the content of the wild parent. The N content of seeds of most progeny fell between the parents but one, 1921 had a higher N content than the wild parent. There was substantial transgressive segregation of S content in the progeny, mostly below the level of the domesticated parent. However positive transgressive segregation was observed for the seed S-content in 1923. We evaluated the ratio of C/N as well because it is better correlated with protein content than N. The C/N ratio (lower value correlated with more protein) of the progeny was likely between that of the parents. However, progenies 1921 and 1923 were similar to the wild parent indicating that these progenies inherited the high-seed protein content of the wild parent. The N/S ratio may indicate improved seed composition. PI424025B had a slightly lower N/S (more S) that NC-Raleigh,

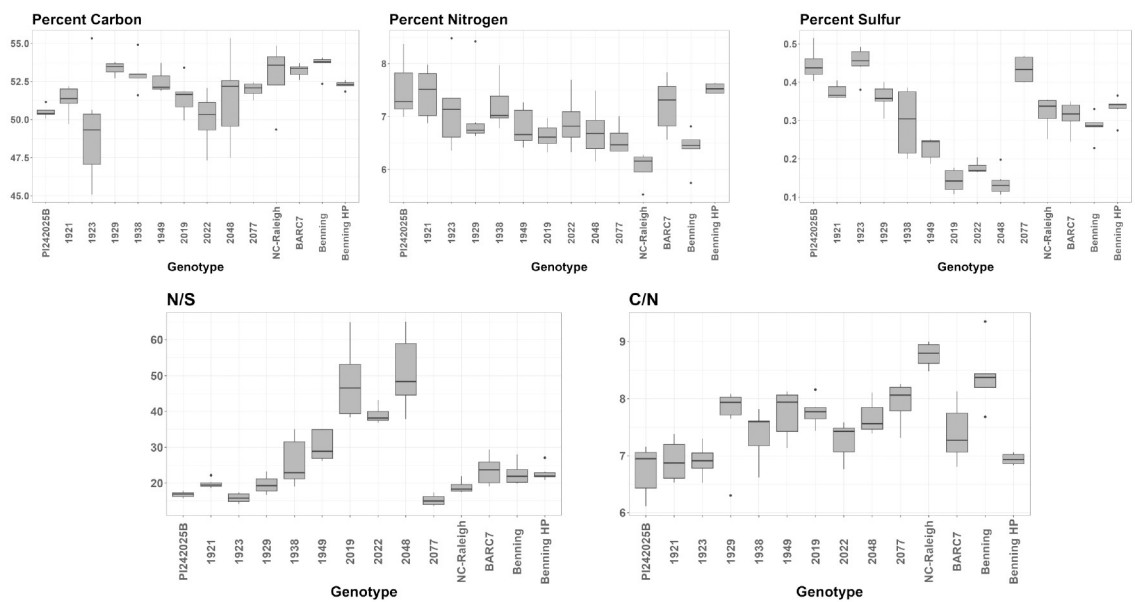

**Fig 3. A box plot showing the %Carbon, %Nitrogen, % sulfur, C/N and N/S for PI424025B, NC-Raleigh and selected progeny.** Also included are the high-seed protein lines BARC-7, Benning HP with the Danbaekkong allele. Benning is the recurrent low-protein parent of Benning HP.

there was substantial transgressive segregation for this trait both better and worse than either parent. Progenies 1923 and 2077 have superior (lower) N/S ratios than the wild parent.

We also evaluated some high protein lines with well characterized high-seed protein phenotypes (Fig 3). Benning is the recurrent low-seed protein parent of Benning HP. The high-protein allele on Benning HP is the Danbaekkong allele on chr20. NC-Raleigh, BARC7, Benning and Benning HP all have a C content greater than PI424025B. However, the C content of Benning HP is lower than Benning. The N content of NC-Raleigh and Benning is inferior to PI424025, but the high-seed protein BARC7 and Benning HP are comparable to PI424025B. Only the C/N of Benning HP is comparable to PI424025B. The S content of PI424025B is superior to all high-protein soybean lines. There is a significant increase in S in Benning HP compared to Benning. However, the Ratio of N/S is comparable between Benning and Benning HP. The N/S ratio of seed from the wild parent is superior to BARC7 and Benning HP. These data indicate that progeny of the wild PI424025B have alleles to elevate protein content and improve S content in soybean seeds relative to other high seed-protein alleles.

Genetic marker analyses were done on the NC-Raleigh, PI424025B and their progeny using the 6K chip [29]. We evaluated linkage of markers to C, N, S, C/N and N/S of seeds. S3 Data shows the results of the mapping experiment for all locations. Associations that were not significant across average data were not considered for further analyses. At a significance level of $\alpha = 0.05$ QTL were identified on chr2, chr15, chr16 and chr20. A locus associated with variation in C content in the seed accounts for 12.0% of the variation mapped to chr16. Evaluation of the phenotypes associated with a nearby marker suggests that the effect is due to outliers in the data and is not considered further.

Fig 4 shows traits associated with improved C, N, S, C/S and C/N mapped colocalized with the POWR1 locus on chr20. This allele accounts for about 11.4%, 31%, 24.5%, 24.5% and 41.9% of variation of C, N, S, C/S and C/N of seeds. Fig 5 shows the phenotypes of progeny with the nearby marker (Gm20_30417244_C_T). In all cases the desirable allele comes from

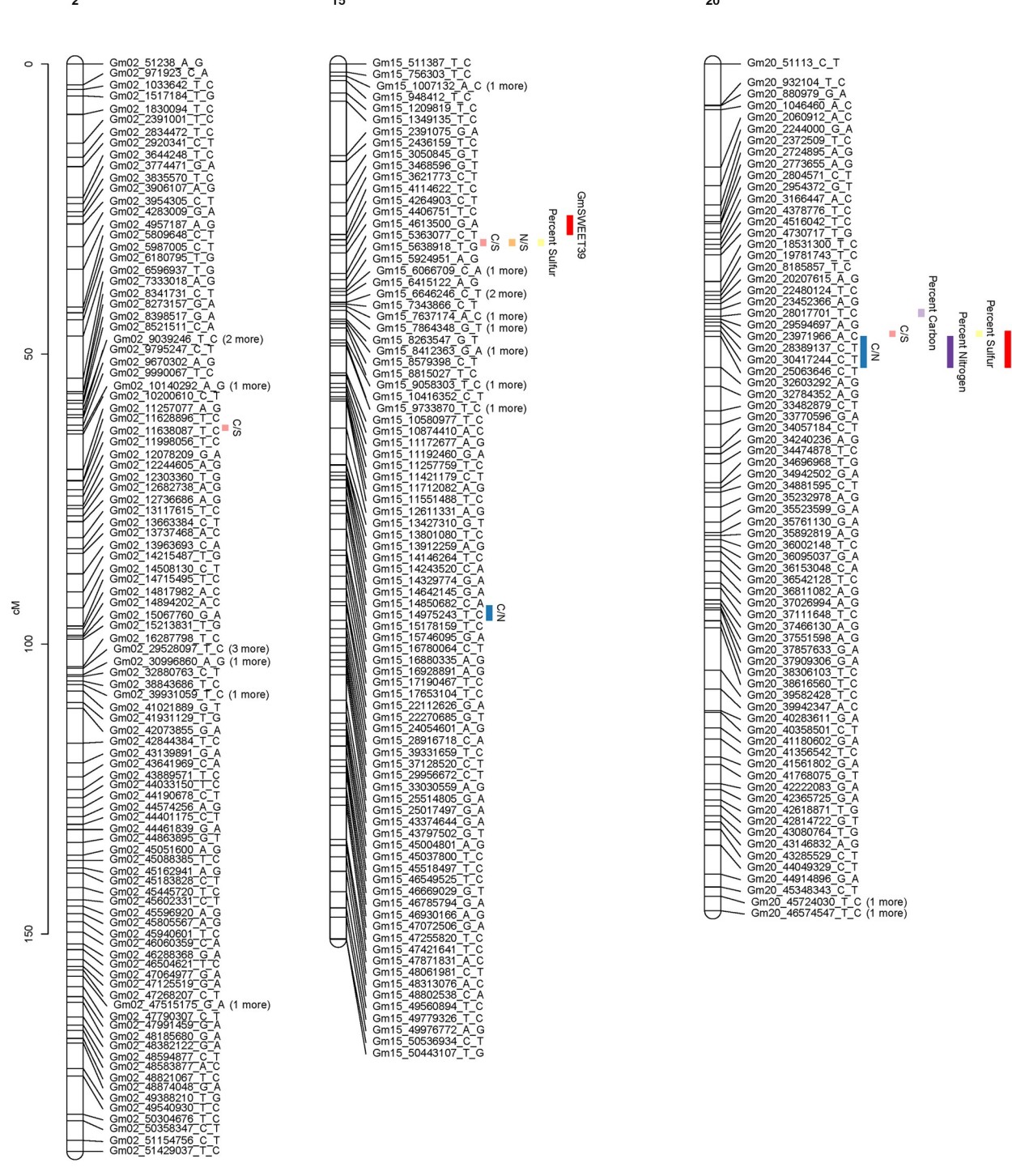

**Fig 4. The map view of QTLs on Chr2, Chr15 and Chr20.** Included are the location of the POWR1 and SWEET39 loci on chr20 and Chr15, respectively.

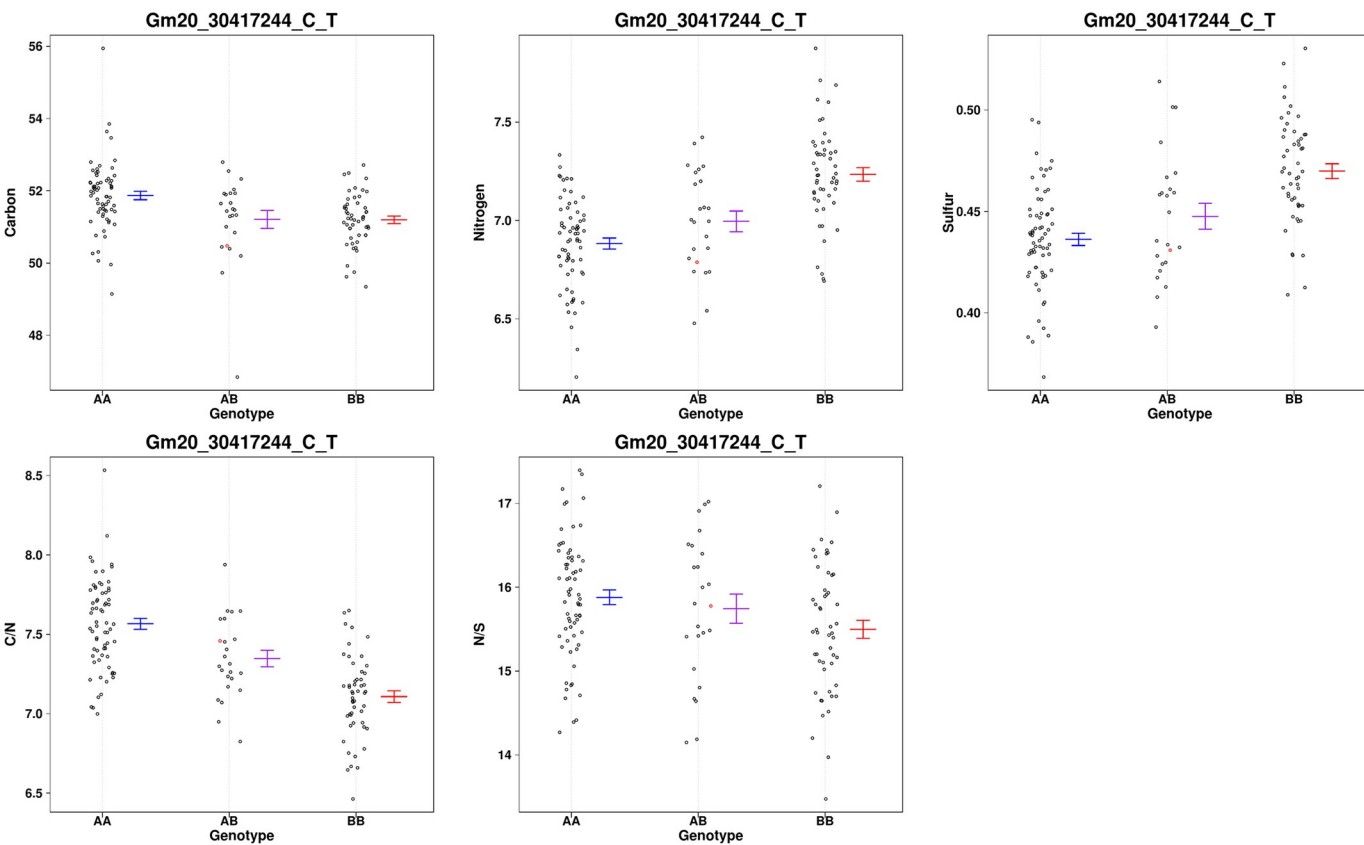

**Fig 5. A scatter plot showing the distribution of %C, %N, %S, C/N and N/S for the marker near the loci identified on chr20.** Missing genotypes are filled in with a single random imputation and plotted in red.

the wild parent (BB). A QTL for the ratio of N to S was not reported for this region even at a significance level of α = 0.1. The difference in the distribution of the N/S shown in Fig 5 must fall below significant levels in this region.

A region on chr15 had a QTL associated with S, C/S and N/S content in the seed near (about 150 Kbp) the GmSWEET39 locus (Fig 4). This QTL is within the interval identified by Zhang et al [23] using higher marker density and two mapping populations. Though ambiguous, it is probable that the S QTL on Chr15 represents SWEET39. Since we are not confident that the S QTL colocalized with SWEET39, we evaluated the gene models between BARC-016533-02084 and BARC-039433-07497 (S4 Data), Interestingly there is a S transporter (Glyma.15G052000) in that region. The phenotype associated with this chr15 locus was distinct from the chr20 locus because only S and C/S and N/S mapped to this locus accounting for 9.8%, 13.1% and 13.2% of the variation, respectively. The relevant phenotypes associated with marker Gm15_4264903_C_T are shown in Fig 6. The apparent difference between C, N and C/N must be insignificant and does not map to the locus even at significance values of 0.1. Fig 6 shows that the high S allele and improved N/S allele are inherited from the wild parent. In chr15 a second allele associated with variation of C/N maps 70 cM from the GmSWEET39 QTL. It accounts for about 11.4% of the variation in this trait. Fig 7 shows the effect Gm15_24054601_A_G has on C, N and C/N. While N content appears different between the allele combinations, it falls below significance even at less significant levels. However, the ratio

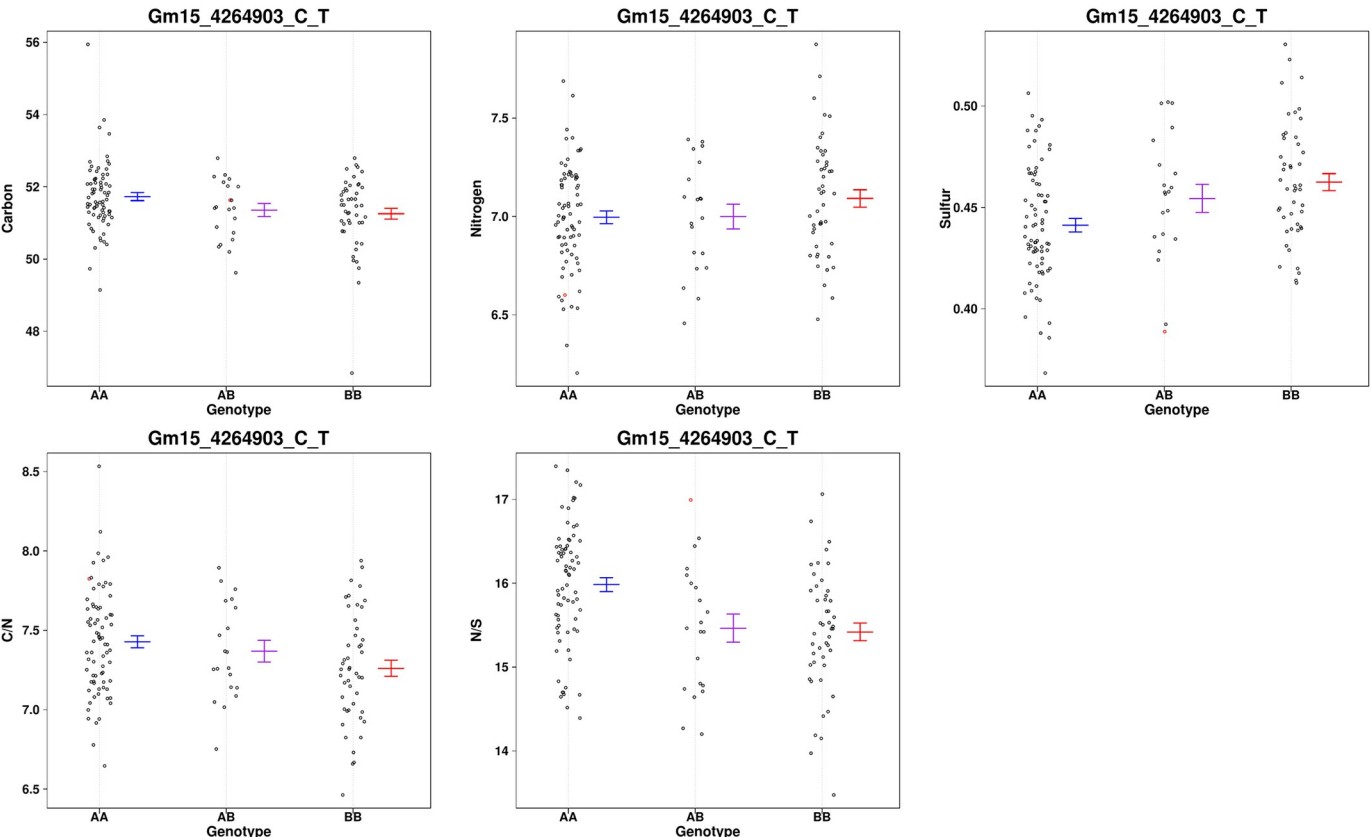

**Fig 6. A scatter plot showing the distribution of %C, %N, %S, C/N and N/S for the marker near the loci identified on chr15 near the SWEET39 locus.** Missing genotypes are filled in with a single random imputation and plotted in red.

of C/N confirmed to be highly correlated with protein content in this population is significant. We evaluated the gene models between BARC-062397-17769—BARC-064209-18584, the markers that flank the QTL (S5 Data). There are many gene models with unknown functions and no obvious candidates to explain the phenotype.

A region on chr2 that accounts for 7.6% of the variation in C/S. Evaluation of nearby marker (Fig 8) Gm02_9990067_T_C indicates the wild allele is associated with elevated S content compared to C. We evaluated the gene models between BARC-016079-02059—BARC-030665-06919 on Chr2 (S6 Data) but no likely explanation of the phenotype was apparent.

## Discussion

We evaluated the progeny of the *G. soja* PI 424025B and NC-Raleigh to determine if this wild PI is a genetic resource that can provide alleles to improve seed-protein content and seed-sulfur content. PI424025B was chosen as the wild parent because it has high-seed protein content, a modest elevation of cysteine in the seed and an elevated amount of S associated with the increase in cysteine [26, 27]. C, N and S content of seeds were measured. The correlation between N and N/C and protein content measured by NIR in the harvest from one NC field site confirms that the elemental analyses accurately predict the protein content. NIR analysis also confirmed that these progenies are lower in seed oil than NC-Raleigh. PI424025B has lower C content than Benning HP with the Danbaekkong allele, Benning, NC-Raleigh and

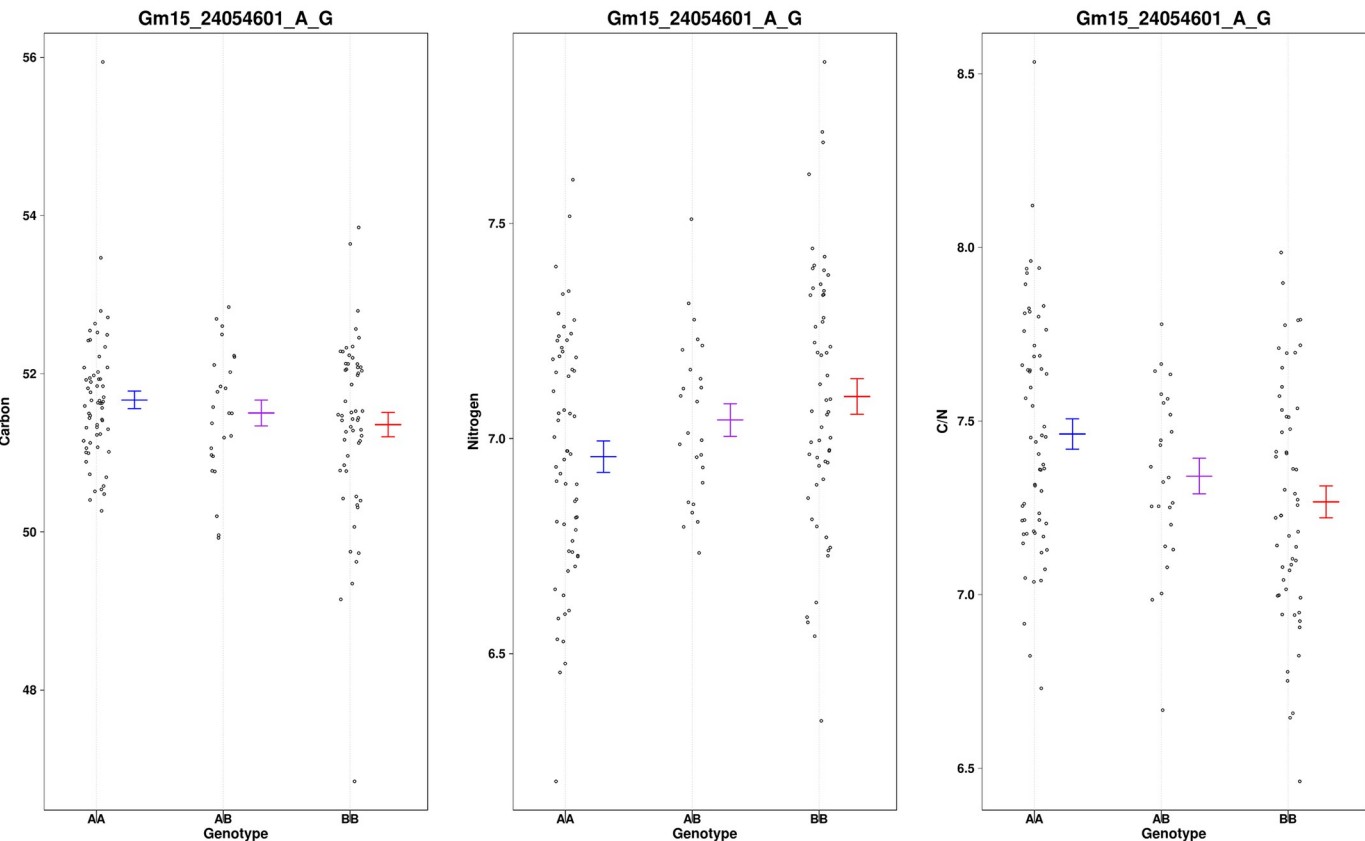

**Fig 7. A scatter plot showing the distribution of C/N for the marker near the loci identified on chr15 approximately 70 cM from the SWEET39 locus.**
Missing genotypes are filled in with a single random imputation and plotted in red.

BARC-7. Additionally, the C content of Benning HP is lower than Benning. This may indicate that the high-protein phenotype is the replacement of C with N, perhaps reflecting a relative increase in oil. N and C/N levels are different between Benning and Benning HP consistent with the expected elevation of seed protein in Benning HP. The high-seed protein genotypes all have increased N content and lower C/N (more N relative to C) compared to the lower-seed-protein genotypes. There is evidence of transgressive segregation for S and N/S content in the lines selected from the mapping population. The principal component analysis of the phenotypic data also supports transgressive segregation of the ratio of N/S because the variation of the progeny into the second dimension is greater than the parents. Kastoori [44] made the same observation in 3 mapping populations of Williams82 crossed with DSR-173, NKS19-90 or Vinton 81. Benning HP seeds had higher S content than Benning seeds, but the N/S ratio was comparable between the two genotypes. A similar N/S ratio between Benning and Benning HP suggests that the increase in S is proportional to the increase in N and does not represent a relative increase in S. PI424025B and some of its progeny have superior N/S content in the seed. Based on these data we anticipate that PI424025B and its progeny will be a source of elevated S-containing amino acids in seed protein. While the heritability of %S and S/C were less than 0.34, in contrast the heritability of N/S was quite high at 0.89. High heritability is valuable for breeding the trait into a superior cultivar.

We mapped the location of QTL associated with C, N, S, C/N and N/S in PI424025, NC-Raleigh and their progeny. QTL associated with S, N, C/N seed content co-segregated with the

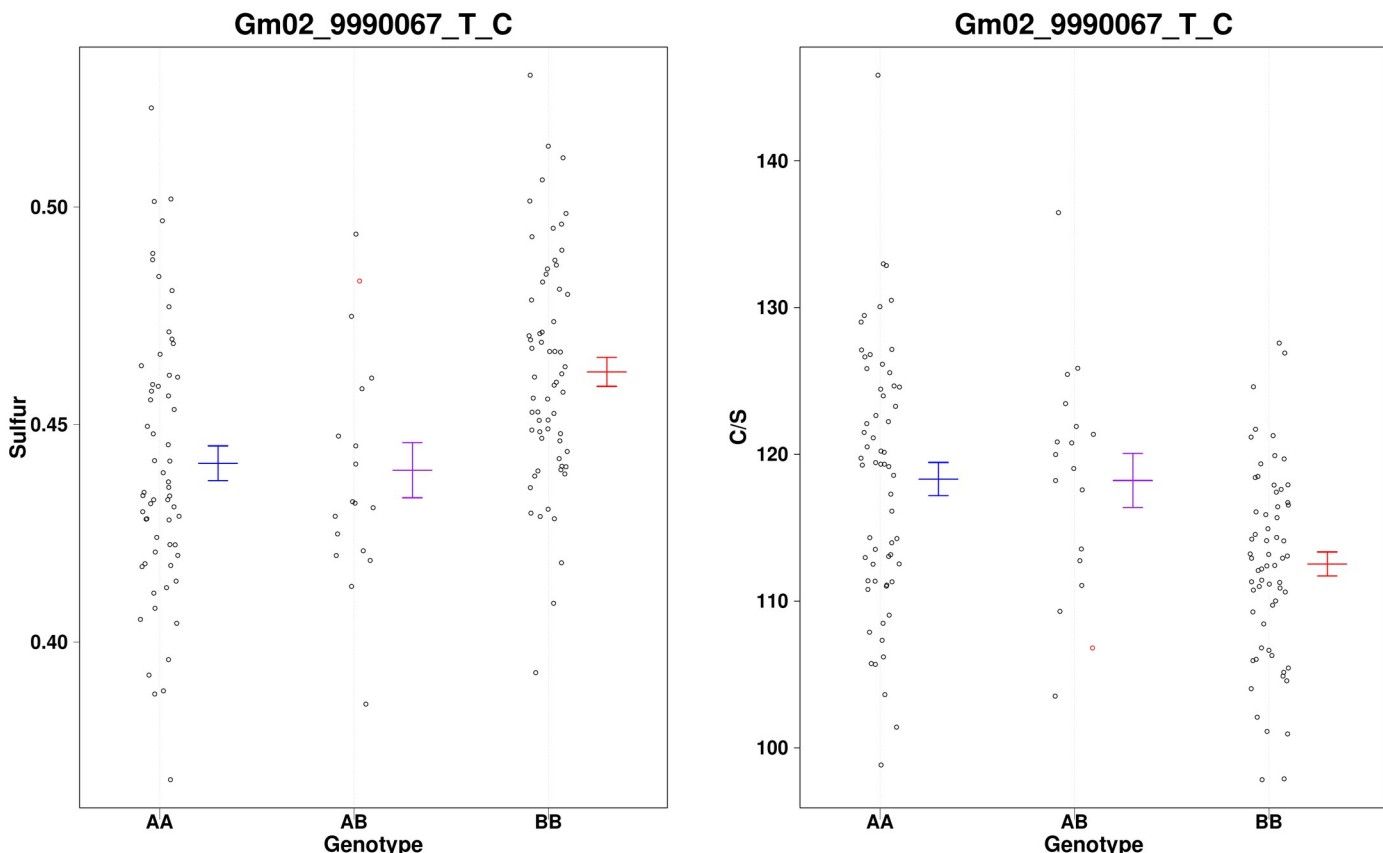

**Fig 8. A scatter plot showing the distribution of %S and C/S for the marker near the loci identified on chr2.** Missing genotypes are filled in with a single random imputation and plotted in red.

well characterized locus on chr20 affecting seed protein. The wild parent contributed the beneficial allele accounting for 12%-31% of the variation. QTL associated with S and N/S seed content are located on chr15. It is unclear if this locus represents the SWEET39 gene or is nearby. These QTL impact the N/S unlike the chr20 QTL, indicating this allele may improve the S content relative to N. Alleles that improve S relative to N are contributed by the wild parent. There is a second QTL on chr15 that impacts the C/N ratio. Again, the high N allele is contributed by the wild parent. A QTL on chr2 from the wild parent is associated with increase in S relative to C. Evaluating this QTL in advanced crosses will indicate if this locus is associated with improved seed S content in advanced progeny. Taken together, PI424025B and its progeny offer a genetic resource to improve soybean seed composition. The value of resources for genetic diversity can be categorized according to their readiness to be incorporated into elite breeding programs [16]. While genetically more diverse than its domesticated relative, wild soybean has numerous negative agronomic properties that make it unsuitable for inclusion into most breeding programs [45]. Identification of individual progeny derived from crosses *G. max* and *G. soja* that inherit wild alleles underpinning improved seed composition of the wild parent but have only half of the wild genome by pedigree is the first step in identifying a bridge between the wild germplasm and elite soybean cultivars [16]. For example, progeny 1923 inherits the high seed-protein and high seed-S content of the wild parent and should be a valuable resource for a second breeding cycle.

## Supporting information

**S1 Data. Percent C,N and S values for NC-Raleigh, PI424025B and their progenies.**
(CSV)

**S2 Data. Genetic marker data for NC-Raleigh, PI424025 and their progenies.**
(CSV)

**S3 Data. A table showing the QTLs at a significance level of α = 0.05 for average of all locations for phenotypes %C, %N, %S and the ratios of these elements.** The chromosome number, pvalue, lod, consensus markers, consensus interval, left marker, right marker and closest marker are reported.
(TIFF)

**S4 Data. Gene models on chr15 between BARC-016533-02084 and BARC-039433-07497 on WM82.a2 assembly.**
(CSV)

**S5 Data. Gene models on chr15 between BARC-062397-17769—BARC-064209-18584 on WM82.a2 assembly.**
(CSV)

**S6 Data. Gene models on chr2 between BARC-016079-02059—BARC-030665-06919 on WM82.a2 assembly.**
(CSV)

## Acknowledgments

We would like to thank the staff at the Red River Research Station in Bossier City, LA for managing field trials and sharing data. We thank the support staff of the USDA-ARS Soybean and Nitrogen Fixation Unit for assisting with this research. We would like to acknowledge the contributions of William "Bill" Todd Molin (1949–2022) to the management of the field trials in Stoneville, MS and extend our gratitude for his eternal curiosity, kind heart, and commitment to research. Mention of trade names or commercial products in this publication is solely for the purpose of providing information and does not imply recommendation or endorsement by the U.S. Department of Agriculture. USDA is an equal opportunity provider and employer.

## Author Contributions

**Conceptualization:** Earl Taliercio.

**Data curation:** Earl Taliercio, Lisa Woodruff.

**Formal analysis:** Jay Gillenwater.

**Investigation:** Lisa Woodruff, Ben Fallen.

**Methodology:** Ben Fallen.

**Resources:** Earl Taliercio, Lisa Woodruff.

**Writing – original draft:** Earl Taliercio, Jay Gillenwater, Ben Fallen.

**Writing – review & editing:** Earl Taliercio.

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
