## [Decision Letter · Decision Letter 0]

4 Jul 2024

PONE-D-24-24053Glycine soja, PI424025, is a valuable genetic resource to improve soybean seed-protein content and compositionPLOS ONE

Dear Dr. Taliercio,

Thank you for submitting your manuscript to PLOS ONE. After careful consideration, we feel that it has merit but does not fully meet PLOS ONE’s publication criteria as it currently stands. Therefore, we invite you to submit a revised version of the manuscript that addresses the points raised during the review process.

We look forward to receiving your revised manuscript.

Kind regards,

Santosh Gudi

Academic Editor

PLOS ONE

Journal Requirements:

"We would like to thank the staff at the Red River Research Station in Bossier City, LA for managing field trials and sharing data. We thank the support staff of the USDA-ARS Soybean and Nitrogen Fixation Unit for assisting with this research. We would like to acknowledge the contributions of William “Bill” Todd Molin (1949-2022) to the management of the field trials in Stoneville, MS and extend our gratitude for his eternal curiosity, kind heart, and commitment to research. Mention of trade names or commercial products in this publication is solely for the purpose of providing information and does not imply recommendation or endorsement by the U.S. Department of Agriculture. USDA is an equal opportunity provider and employer. This research was in part funded by the United Soybean Board"

"U.S. Department of Agriculture. USDA is an equal opportunity provider and employer. This research was in part funded by the United Soybean Board "

Additional Editor Comments:

It is obvious that QTL mapping will identify several QTLs for the studied trait. However, pinpointing the strong QTLs with major effect on phenotype will help the breeders to transfer beneficial traits in breeding lines. So, authors need to pinpoint major QTLs with strong effect on phenotype. Additionally, soybean have the well annotated genome. Please do candidate gene analysis from the major QTLs regions.

Though it is good work, authors failed to discuss the obtained results appropriately. Therefore, results section needs to be improved.

Introduction and Discussions are the key sections for MS as they will give brief background about the subject/study and validation for your results, respectively. Improve these sections by citing the most recent references. Such as:

• Singh, G., Gudi, S., Amandeep, Upadhyay, P., Shekhawat, P. K., Nayak, G., ... & Ayoubi, H. (2022). Unlocking the hidden variation from wild repository for accelerating genetic gain in legumes. Frontiers in plant science, 13, 1035878.

• Gudi, S., Saini, D. K., Singh, G., Halladakeri, P., Shamshad, M., Tanin, M. J., ... & Sharma, A. (2021). Unravelling consensus genomic regions associated with quality traits in wheat (Triticum aestivum L.) using meta-analysis of quantitative trait loci. bioRxiv, 2021-11.

In material and method “Phenotype data Analysis” section come before “Molecular marker and QTL analysis”

Change title from “conclusion” to “discussion”

Reviewers' comments:

Reviewer's Responses to Questions

**Comments to the Author**

1. Is the manuscript technically sound, and do the data support the conclusions?

Reviewer #1: Partly

Reviewer #2: Yes

2. Has the statistical analysis been performed appropriately and rigorously? 

Reviewer #1: No

Reviewer #2: Yes

3. Have the authors made all data underlying the findings in their manuscript fully available?

Reviewer #1: No

Reviewer #2: Yes

4. Is the manuscript presented in an intelligible fashion and written in standard English?

Reviewer #1: No

Reviewer #2: Yes

5. Review Comments to the Author

Reviewer #1: The study attempts to identify sources for superior seed-protein content and composition by generating mapping populations and then unravelling the genetic basis of traits related to elemental composition in seeds. The purpose of the study is important as improving soybean protein content and composition could amount to better nutritional and market value.

Major comments:

While the superior line identified may be possessing higher protein content and desirable alleles for protein content and composition, the study has issues with experimental plan and presentation. Most of the paper focuses on the elemental composition of C, N, S their ratios as proxies for protein content and composition.

1. Writing is not up to the mark – fails to mention key aspects such as why these traits were chosen to be studied in detail are missing and also there are no adequate and appropriate references to support the sentences in the paper at most of the places.

2. Highly Incoherent presentation of the findings and lack of depth in analysing and discussing how the findings in the study correlate with the findings from previous studies.

3. An elaboration on why sulfur is important for protein quality can be added in the introduction. Additionally, a few lines on why carbon partitioning and allocation during maturity stages is important for protein content should be mentioned here for clearer understanding of the readers.

4. In the introduction, the penultimate paragraph says that representation of QTL by the said markers is ambiguous. The physical position of the markers and the POWR1 locus can be obtained and checked for colocalization. This sentence can be changed in the manuscript.

5. The authors have not adequately mentioned or explained in this paper why NC Raleigh was chosen as a parent for crossing and generation of mapping population with PI424025B.

6. The authors must use the full form of abbreviations the first time they use it in the paper and use the abbreviated form during the subsequent uses.

7. What was the area of the plots used for the experiment?

8. As it has been mentioned that enough seeds were not available for protein estimation using NIR, the authors need to mention the quantity of seeds required for protein and oil measurement using NIR, and what quantity was actually used (for one location).

9. “Molecular marker and QTL analysis” section in Materials and methods can be merged with “Linkage map construction” part for more coherence.

10. More insights could be obtained by analysing the phenotype data in more detail. Descriptive stats, ANOVA and biplots to understand the GxE interactions across the 5 locations can be done. Also, assessing some agronomic traits would be more informative.

11. In the “Linkage map construction” section in Materials and methods, I suggest the authors to check the collinearity of the linkage map with the physical map using the physical positions of the 6K panel.

12. Since protein was not estimated at all the locations but N was, is there a formula to convert the amount of N to %protein across all locations?

13. Reference missing for the sentence - “The ratio of N to C is often reported which gave a better correlation of (0.78) with seed protein measured by NIR.”

14. The reduction in carbon source towards the terminal stages of the plant may contribute to lower protein content due to competition for carbon skeleton with oil. How does the finding in your study correlate with previous studies? Since NC Raleigh has high C content, how does that fit here since it has low protein content? Interpretations of your results must be done thoroughly by consulting previous similar studies.

15. The GmSWEET39 gene is 388KB upstream of the left marker of the QTL (SoyBase, Glyma2.0). It may be better not to consider that the QTLs on chr15 and GmSWEET39 are colocalized as mentioned in the paper. Claims must be thoroughly checked for accuracy before presenting.

16. Candidate gene analyses could be performed on the QTLs obtained to see what kind of genes present/possible molecular mechanisms are.

17. Protein content and oil have not been estimated in all but one of the locations used in the study. It is unclear as to how the C, N, S content translate to protein and oil content. Correlation alone cannot be used as a metric for using the C, N, S compositions as proxy for protein content and oil performance across different locations. Overall, the study lacks detail in discussion and analyses. Due to these reasons, the study does not have adequate merits for publication currently.

Reviewer #2: In this study authors have developed a Recombinant inbred line (RIL) population using Glycine soja as one of the parent. Phenotyping of the RILs was done for seed protein, carbon, nitrogen and sulphur content. Linkage map was developed using SNP data and new QTLs for high protein, carbon, nitrogen and sulphur content were identified were. Overall, this manuscript provides new information regarding the seed composition traits in soybean. Following points may be considered:

1. Additional information regarding candidate genes and QTLs may be provided.

2. Application of the outcome may be discussed in more detail.

3. Recently published articles related to the field may be cited and briefly discussed.

6. PLOS authors have the option to publish the peer review history of their article (what does this mean?). If published, this will include your full peer review and any attached files.

Reviewer #1: No

Reviewer #2: No

---

## [Author Response · Author response to Decision Letter 0]

19 Aug 2024

Removed funding information from acknowledgments 

 We have removed financial information from acknowledgments

Amended financial statement is included in cover letter

Phenotypic and marker data has been added to supplemental information

Add references

 We have added and updated numerous references

Provide data

 We have added phenotypic and marker data to supplementary material. 

Additional Editor Comments:

It is obvious that QTL mapping will identify several QTLs for the studied trait. However, pinpointing the strong QTLs with major effect on phenotype will help the breeders to transfer beneficial traits in breeding lines. So, authors need to pinpoint major QTLs with strong effect on phenotype. Additionally, soybean have the well annotated genome. Please do candidate gene analysis from the major QTLs regions.

We have added a discussion of candidate genes. 

Though it is good work, authors failed to discuss the obtained results appropriately. Therefore, results section needs to be improved.

Introduction and Discussions are the key sections for MS as they will give brief background about the subject/study and validation for your results, respectively. Improve these sections by citing the most recent references. Such as:

• Singh, G., Gudi, S., Amandeep, Upadhyay, P., Shekhawat, P. K., Nayak, G., ... & Ayoubi, H. (2022). Unlocking the hidden variation from wild repository for accelerating genetic gain in legumes. Frontiers in plant science, 13, 1035878.

• Gudi, S., Saini, D. K., Singh, G., Halladakeri, P., Shamshad, M., Tanin, M. J., ... & Sharma, A. (2021). Unravelling consensus genomic regions associated with quality traits in wheat (Triticum aestivum L.) using meta-analysis of quantitative trait loci. bioRxiv, 2021-11.

In material and method “Phenotype data Analysis” section come before “Molecular marker and QTL analysis”

We have added Unlocking the hidden variation from wild repository for accelerating genetic gain in legumes 

Change title from “conclusion” to “discussion”

 Change has been made

Reviewer 1. 

1. Writing is not up to the mark – fails to mention key aspects such as why these traits were chosen to be studied in detail are missing and also there are no adequate and appropriate references to support the sentences in the paper at most of the places.

Response: added reference (Soy.xlsx (live.com) to value of oil in introduction. 

I added references for the for the effort put into analysis of seed-protein content in soybean including more references in the first paragraph of the introduction.

I added references for the negative correlation between seed protein content and yield in the first paragraph of the introduction.

Added reference for USDA-N7007 which is a 2024 publication

2. Highly Incoherent presentation of the findings and lack of depth in analyzing and discussing how the findings in the study correlate with the findings from previous studies.

Response: I have added a more detailed analysis of the phenotypic data that includes heritability in results. 

3. An elaboration on why sulfur is important for protein quality can be added in the introduction. Additionally, a few lines on why carbon partitioning and allocation during maturity stages is important for protein content should be mentioned here for clearer understanding of the readers.

Response:

Added to the 3rd paragraph of the introduction: 

Feed for animals is the most important market for soy meal (https://mosoy.org/about-soybeans/soybean-uses/soybeans-as-feed/). While an excellent source of protein soymeal is low in the sulfur-containing amino acids (George & De Lumen, 1991).

4. In the introduction, the penultimate paragraph says that representation of QTL by the said markers is ambiguous. The physical position of the markers and the POWR1 locus can be obtained and checked for colocalization. This sentence can be changed in the manuscript.

Response

 I don’t think it is possible to be more confident about the colocalization of “high cysteine traits” in soybase with the POWR 1 locus because ANOVA of nearby markers were used to estimate the differences in seed-cysteine content. In fact soybase warns against extrapolating actual distances. Part of the warning soybase gives follows. “Very importantly, since the gene underlying the QTL may only be loosely linked to the marker tested it could be anywhere +/- 0 – 30 cM in either direction from QTL position shown in genetic map” QTL to Nearest Sequence Feature Search Results (soybase.org) . 

5. The authors have not adequately mentioned or explained in this paper why NC Raleigh was chosen as a parent for crossing and generation of mapping population with PI424025B.

Response:

Added to plant material

“NC-Raleigh was chosen as a parent because its flowering period overlapped with PI424025B , it is lodging resistant and is resistant to frogeye leaf spot (Burton, Carter, Fountain, & Bowman, 2006)”

6. The authors must use the full form of abbreviations the first time they use it in the paper and use the abbreviated form during the subsequent uses.

We have tracked corrections

7. What was the area of the plots used for the experiment?

Response:

Amended to “Seeds were planted in five-foot single row plots separated by two-foot alleys.”

8. As it has been mentioned that enough seeds were not available for protein estimation using NIR, the authors need to mention the quantity of seeds required for protein and oil measurement using NIR, and what quantity was actually used (for one location).

Response: The actual quantity would vary by seed size but the samples were ground, so we added the surface area provided by the dish used

Added “JRS in the large dish (103 cm2 surface area)” to “NIR and analyses of C, N and S 

“ in Materials and Methods, 

9. “Molecular marker and QTL analysis” section in Materials and methods can be merged with “Linkage map construction” part for more coherence.

Response:

Reorganized more logically. Changes are tracked. 

10. More insights could be obtained by analyzing the phenotype data in more detail. Descriptive stats, ANOVA and biplots to understand the GxE interactions across the 5 locations can be done. Also, assessing some agronomic traits would be more informative

Response

This was a good idea. We have added and PCA and ANOVA data including heritability. 

to phenotype analysis added “ Principal Component analysis was performed in R using prcomp and data was centered and scaled. The biplot was generated using the factorextra package (1.0.7) biplot function. ANOVA was performed in R using CAR package. 

We have added FigA to represent PCA and Table A to represent ANOVA . We have added two paragraphs for Results section 

Fig A shows the principal component analysis of the % C, %S, %N, N/C, S/C and N/S from all locations and the average of locations. Dimension 1 and dimension 2 account for a variation of 41% and 15% respectively. The variation of these values is similar across locations. N and N/C contribute substantially to variation and are inversely correlated to %C which lies along the same line but in the opposite direction. N/S contributes to DIM2 and interestingly is not closely correlated with %N or S or S/C. 

ANOVA analysis of phenotypic data is shown in Table A. There is a substantial Environment and GenotypeXEvirionment affect for all phenotypes except the genotypeXEnvirioment effect of %C is insignificant. The broad sense heritability for %S and S/N is low at 0.22 and 0.34 respectively. However, the H2 for N/S is high at 0.89. Broad sense heritability for %N and N/C is 0.73 and 0.85, respectively. 

We also added mention of the PCA and ANOVA analyses to the discussion

“The principal component analysis of the phenotypic data also supports transgressive segregation of the ratio of N/S because the variation of the progeny into the second dimension is greater than the parents.’

‘While the heritability of %S and S/C were less than 0.34, in contrast the heritability of N/S was quite high at 0.89. High heritability is valuable for breeding the trait into a superior cultivar. ‘

11. In the “Linkage map construction” section in Materials and methods, I suggest the authors to check the collinearity of the linkage map with the physical map using the physical positions of the 6K panel

Response:

We indicate the number () of markers that were eliminated because they did not group to expected linkage maps. We clarified the relationship of the sweet39 and the close by qtl from this paper. 

12. Since protein was not estimated at all the locations but N was, is there a formula to convert the amount of N to %protein across all location

Response

Added “It is possible to estimate protein content from N content (Krul, 2019). Typically an N to protein conversion factor of 6.25 is used but there is debate about the best value to use (Krul, 2019).” 

13. Reference missing for the sentence - “The ratio of N to C is often reported which gave a better correlation of (0.78) with seed protein measured by NIR.”

To first paragraph in results added. “Also included were ratios of N to C (N/C), S to C (S/C) and nitrogen to sulfur (N/S). Ratios are often included in data from elemental analyses (Dhanapal et al., 2015; Ning, Yang, Li, & Fritschi, 2018; Peng, Li, & Fritschi, 2014). Significant correlation of shoot N/C with QTL were identified in a panel of soybean lines (Dhanapal et al., 2015)’

14. The reduction in carbon source towards the terminal stages of the plant may contribute to lower protein content due to competition for carbon skeleton with oil. How does the finding in your study correlate with previous studies? Since NC Raleigh has high C content, how does that fit here since it has low protein content? Interpretations of your results must be done thoroughly by consulting previous similar studies

Response: 

I don’t Think these data can really address this question. The addition of the PCA in figure one does suggest that there is a negative correlation between C and N/C which is consistent with the competition the reviewer describes. 

15. The GmSWEET39 gene is 388KB upstream of the left marker of the QTL (SoyBase, Glyma2.0). It may be better not to consider that the QTLs on chr15 and GmSWEET39 are colocalized as mentioned in the paper. Claims must be thoroughly checked for accuracy before presenting.

Response: We changed the call to ambiguous because, as the reviewer points out, SWEET39 does not lie between the flanking markers. However, the region flanking our markers (BARC-016533-02084 - BARC-039433-07497) is completely encompassed by the interval described in the SWEEY39 reference. 

We added “ Our QTL is within the interval identified by Zhang et al using higher marker density and two mapping populations. Though ambiguous it is probable that the S QTL on Chr15 represents SWEET39. Since we are not confident that the S QTL colocalized with SWEET39 we evaluated the gene models between BARC-016533-02084 and BARC-039433-07497 (Supplementary data A), Interestingly there is a S transporter () in that region.”

16. Candidate gene analyses could be performed on the QTLs obtained to see what kind of genes present/possible molecular mechanisms are

See the response to question 15

We added

Materials and methods:

To identify genes in the region of QTL, we identified the sequences of the right and left markers (shown in supplementary data 1) in the Wm82.a2 assembly using the BLAST tool at soybase. The tool in the soybase genome browser was used to download the gene models in the targeted region and the soybase annotation tool was used to download the annotation

Results:

We evaluated the gene models between BARC-062397-17769 - BARC-064209-18584, the markers that flank the QTL (supplementary data 3). There are a large number of gene models with unknown functions but no obvious candidates to explain the phenotype. 

We evaluated the gene models between BARC-016079-02059 - BARC-030665-06919 on Chr2 (supplementary data 4) but no likely explanation of the phenotype was apparent. 

17. Protein content and oil have not been estimated in all but one of the locations used in the study. It is unclear as to how the C, N, S content translate to protein and oil content. Correlation alone cannot be used as a metric for using the C, N, S compositions as proxy for protein content and oil performance across different locations. Overall, the study lacks detail in discussion and analyses. Due to these reasons, the study does not have adequate merits for publication currently.

It is true that we have not definitively determined that oil is negatively correlated with seed-protein because we are only evaluating one environment. We are not proposing anything extraordinary because the negative correlation between seed-oil and seed-protein is well documented. This will be information other breeders want to decide if these progenies will fit into their breeding program. I think it is worth including. I have added documentation per the reviewers request to support the correlation between N content and protein. The higher heritability of the N/S ratio compared to %S could turn out to be a valuable tool for breeders. It would have been desirable to measure S-containing amino acids but the cost was absolutely prohibitive. The increased detail in the form of more and recent references and improved analyses adding the PCA and ANOVA analysis address the reviewers concerns. 

Reviewer 2:

1. Additional information regarding candidate genes and QTLs may be provided.

See response to reviewer 1 questions 15 and 16.

2. Application of the outcome may be discussed in more detail.

Revised discussion

(Sanchez et al., 2023). While genetically more diverse than its domesticated relative, wild soybean has numerous negative agronomic properties that make it unsuitable for inclusion into most breeding programs (Hyten et al., 2006). Identification of individual progeny derived from crosses G. max and G. soja that inherit wild alleles underpinning improved seed composition of the wild parent but have only half of the wild genome by pedigree is the first step in identifying a bridge between the wild germplasm and elite soybean cultivars(Sanchez et al., 2023). For example, progeny 1923 inherits the high seed-protein and high seed-S content of the wild parent and should be a valuable resource for a second breeding cycle.

2. Recently published articles related to the field may be cited and briefly discussed

I have added new and updated references

---

## [Decision Letter · Decision Letter 1]

4 Sep 2024

Glycine soja, PI424025, is a valuable genetic resource to improve soybean seed-protein content and composition

PONE-D-24-24053R1

Dear Dr. Taliercio,

We’re pleased to inform you that your manuscript has been judged scientifically suitable for publication and will be formally accepted for publication once it meets all outstanding technical requirements.

Kind regards,

Santosh Gudi

Academic Editor

PLOS ONE

Additional Editor Comments (optional):

Reviewers' comments:

Reviewer's Responses to Questions

**Comments to the Author**

1. If the authors have adequately addressed your comments raised in a previous round of review and you feel that this manuscript is now acceptable for publication, you may indicate that here to bypass the “Comments to the Author” section, enter your conflict of interest statement in the “Confidential to Editor” section, and submit your "Accept" recommendation.

Reviewer #1: (No Response)

Reviewer #2: All comments have been addressed

2. Is the manuscript technically sound, and do the data support the conclusions?

Reviewer #1: (No Response)

Reviewer #2: Yes

3. Has the statistical analysis been performed appropriately and rigorously? 

Reviewer #1: (No Response)

Reviewer #2: Yes

4. Have the authors made all data underlying the findings in their manuscript fully available?

Reviewer #1: (No Response)

Reviewer #2: Yes

5. Is the manuscript presented in an intelligible fashion and written in standard English?

Reviewer #1: (No Response)

Reviewer #2: Yes

6. Review Comments to the Author

Reviewer #1: (No Response)

Reviewer #2: The revised manuscript is of high scientific quality and provides important information in this field.

7. PLOS authors have the option to publish the peer review history of their article (what does this mean?). If published, this will include your full peer review and any attached files.

Reviewer #1: No

Reviewer #2: No

---

## [Editor Report · Acceptance letter]

6 Sep 2024

PONE-D-24-24053R1 

PLOS ONE

Dear Dr. Taliercio, 

I'm pleased to inform you that your manuscript has been deemed suitable for publication in PLOS ONE. Congratulations! Your manuscript is now being handed over to our production team.

Kind regards, 

on behalf of

Dr. Santosh Gudi 

Academic Editor

PLOS ONE